# Covid-19 and the quality of life of people with dementia and their carers—The TFD-C19 study

Stephanie Daley [1]*, Nicolas Farina[1], Laura Hughes[1], Elise Armsby[2], Nazire Akarsu[2], Joanna Pooley[2], Georgia Towson[2], Yvonne Feeney[1], Naji Tabet[1], Bethany Fine[2], Sube Banerjee[3]

1 Centre for Dementia Studies, Brighton and Sussex Medical School, University of Sussex, Falmer, East Sussex, United Kingdom, 2 Research and Development Department, Sussex Partnership NHS Foundation Trust, Hove, East Sussex, United Kingdom, 3 Faculty of Health, University of Plymouth, Plymouth, Devon, United Kingdom

☯ These authors contributed equally to this work.
* s.daley@bsms.ac.uk

**Data Availability Statement:** All relevant data are within the manuscript.

**Funding:** This study was funded by the National Institute for Health Research (NIHR) Applied

## Abstract

### Introduction

COVID-19 has placed unprecedented pressure on dementia health and social care systems worldwide. This has resulted in reduced services and support for people with dementia and their family carers. There are gaps in the evidence on the impact of the pandemic on Quality of Life (QoL). We carried out a study on the impact of the pandemic on the QoL of a group of people with dementia and their family carers who were part of a larger existing cohort study.

### Methods

We quantitatively measured QoL, on two occasions during the two national lockdowns in 2020 and compared these data with those obtained when they entered the study (before the pandemic). Measures used included: DEMQOL-Proxy, Clinical Dementia Rating Scale and C-DEMQOL. To understand how QoL changed over time, a repeated measures ANOVA was run for each dependent variable with the following variables entered as co-variates: duration in study, baseline dementia severity, gender of the family carer, gender of the person with dementia, family carer relationship, dementia type, living status, age of the person with dementia, and age of the family carer.

### Results

248 participants took part in the study. QoL scores did not significantly decline between either time period for the person with dementia or their family carer. There was variation in subgroups; with co-resident status, carer relationship, gender of the person with dementia, age of the person with dementia, and baseline cognitive status influencing QoL outcomes in family carers.

Research Collaboration Kent, Surrey, Sussex (grant number N/A), as well as Sussex Partnership NHS Foundation Trust and Brighton & Sussex Medical School The funders had no role in study design, data collection and analysis, decision to publish, or preparation of the manuscript.

**Competing interests:** The authors have declared that no competing interests exisit.

## Discussion

It is striking that people with dementia and their carers did not report a decline in QoL during the pandemic or in the months following restrictions suggesting the possibility of resilience. Variation in subgroups suggests that specific groups of family carers were more vulnerable to lower QoL; indicating the need for more tailored, nuanced support during this period.

## Introduction

The COVID-19 pandemic has placed unprecedented pressure on health and social care systems worldwide as they respond to the virus. This has resulted in a significant impact on the availability of services and support for people with dementia and their family carers. In the UK, COVID-19 related public health restrictions led to the suspension of most face-to-face social support services, at least temporarily, with social support service usage for people with dementia decreasing significantly since the pandemic [1, 2]. In addition, social isolation and restrictions meant that other sources of informal support, such as non-resident family and social networks, are also compromised. The wider literature suggests that people with long-term conditions are particularly vulnerable during national emergencies, and the World Health Organisation [3] has acknowledged that older adults, particularly those with dementia, may be severely psychologically impacted during the COVID-19 pandemic.

Social distancing restrictions have affected the ability of family carers to access support, attend support groups and has made caring for the person with dementia increasingly difficult, resulting in increased family carer stress levels [4]. A UK survey of 569 older adults and people affected by dementia found that a reduction in social support services received during the pandemic was significantly associated with reduced levels of wellbeing in family carers [1]. An online study in July 2020 of family carers of people with dementia described increased responsibilities, strain on the caring relationship, and feelings of exhaustion and guilt [5] It has been reported that 73% of family carers experienced an increase in caring responsibilities in lockdown, largely due to worsening dementia symptoms in the person they care for, with 95% of respondents reporting that the increase in caring duties had a negative impact on their mental and/or physical health [6]. This included feeling exhausted, anxious, depressed, and having problems sleeping. Additionally, disorientation, confusion, and distress in people with dementia has significantly increased anxiety and burden in family carers [7]. A Greek survey of 204 family carers found cognitive, emotional, and physical decline in the people with dementia with a consequently increased carer workload [8]. Similarly, in Argentina the level of burden carers experienced was reported to be higher after four weeks of lockdown, especially when caring for people with advanced stages of dementia [9]

It is clear that while social restrictions were put in place to protect vulnerable people from COVID-19, they have also led to increased burden and strain on family carers of people with dementia, with a negative impact on their mental health. However, there remain important gaps in the evidence base particularly in terms of overall impact as measured by Quality of Life (QoL). QoL can be defined as the subjective assessment of the position of the individual in their life, in the context to their culture and value system, relating to their goals, expectations and concerns, impacted by their physical and psychological health, level of independence, and social relationships [10]. Living with, or caring for somebody with dementia impacts upon all of the factors known to affect QoL, it is likely that Covid-19 and associated restrictions may have been more impactful for this group of people. There is therefore a need to measure the

impact of Covid-19, using condition-specific measures of QoL, which are tailored to the challenges and realities of daily life with dementia. There is also a need for further COVID-19 research in dementia that uses large and well characterised cohorts of people with dementia with measurement of QoL of carers and people with dementia.

We therefore carried out a study of the impact of the pandemic on a group of people with dementia and their family carers who were part of an existing cohort study, the evaluation of the Time for Dementia Programme [11, 12]. Time for Dementia is an undergraduate educational programme in the South-East of England and as part of its evaluation, before the pandemic, the QoL of family carers of people with dementia, as well as the social functioning, QoL, and illness severity of the person with dementia were assessed. In this study (TFD-C19), we followed up this participant group and repeated the same measures during the two national lockdowns in 2020 with the aim of investigating the impact of the COVID-19 pandemic and restrictions compared with existing pre-pandemic data. Our study aim was to understand how COVID-19 had affected the QoL, wellbeing, and care of people with dementia and their family carers.

## Methods

### Study design

TFD-C19 was nested into the Time for Dementia (TFD) study. We quantitatively measured QoL, social functioning and disease severity during the two national lockdowns in 2020 and compared these data with those obtained when they entered the programme (before the pandemic). The study was approved by the NHS Health Research Authority London Queen Square Research Ethics Committee (15/LO/0046).

### Contextual information

On 26 March 2020, lockdown measures were implemented in England. Those considered clinically vulnerable were asked to shield, which included recommendations to stay at home and avoid social contact. Most non-urgent, routine, healthcare appointments were cancelled. On 1 June 2020 easing of restrictions occurred, those that were shielding were allowed outdoors with people living outside of their household. In June, other businesses and non-urgent healthcare appointments began to reopen, although many support services for people with dementia did not reopen. From 13 June 2020 those living alone were allowed to spend time with one other household as part of a "support bubble", without social distancing measures. This new rule did not apply to those who were shielding. On 1 August 2020, those shielding no longer needed to isolate. Measures continued to ease (outside of localised restrictions, not affecting the South East of England) until 5 November 2020 when a second national lockdown was put into place. A third national lockdown was put in place on 5th January 2021.

### Participants and consent

Participants were family carers of people with dementia who were taking part in the TFD study evaluation. The study definition of family carers included married and unmarried partners, children, siblings, extended family and close friends. The term 'family carers' is used since it was preferred by our lived experience advisory group. Due to the difficulty in assessing capacity over the telephone, the people with dementia on the programme were not interviewed for this study. Existing study participants were approached by a research worker. Those who were interested were sent a study information sheet, and verbal consent was given by those who wanted to take part. An appointment was made to collect measures by telephone.

## Study assessments

All carers completed the same battery of instruments at Time 2 (May/June 2020 during the first national lockdown) and Time 3 (October/November 2020 part-way through the second national lockdown) as they had at Time 1 baseline assessment when joining TFD (March 2018 –March 2020).

All measures were completed at each time point, and are described below:

**DEMQOL-Proxy [13].**   A proxy measure of QoL in people with dementia with 31 items. The measure has two domains; functioning and emotions. Scores range from 31–124, with higher scores indicating a better QoL for the person with dementia. Internal consistency for the measure is $\alpha$ = 0.87–0.92, and test-retest reliability is ICC = 0.67–0.84 [14].

**C-DEMQOL [15].**   A measure of QoL in family carers of people with dementia. A 30-item measure with scores ranging from 30–150 with higher scores representing better QoL for family carers. The measure provides a rating for overall QoL, which is made up from five sub-scales: meeting personal needs, carer wellbeing, carer-patient relationship, confidence in the future, and feeling supported. Reliability for the total QOL score, is good; with $\omega$ = 0.97, with a range of 0.82 to 0.95 for each of the five sub-domains.

**The clinical dementia rating scale (CDR) [16].**   A proxy report of cognitive and functional performance in dementia. A semi-structured interview is undertaken to rate cognitive performance in 6 domains, memory, orientation, judgement and problem solving, community affairs, home and hobbies and personal care. The total CDR score provides the stage of disease severity (0 = Normal to 3 = Severe dementia). The CDR Sum of Boxes (SOB) was calculated by summing each of the domain box scores; ranging from 0 to 18 with higher scores representing greater cognitive impairment. Inter-rater reliability for this measure has been reported as $_k$ = 0.53–0.80 [17]. This measure was used to assess whether dementia was associated with change in QoL scores over time.

## Patient and public involvement

People with dementia and their family carers were involved throughout the research process though a study specific dementia advisory group, including designing the study, interpreting findings, and helping to disseminate the intervention and findings.

## Analysis

Summary statistics were generated on demographic information for the person with dementia and family carer. Means and standard deviations were reported for continuous data, frequencies and percentages were reported for categorical data. Categorical data were dichotomised. Total scores for the C-DEMQOL (and its subscales) and DEMQOL-Proxy were calculated according to their original development guidelines. This included the rules for handling missing data.

To understand how QoL changed over time, a repeated measures ANOVA was run for each dependent variable with the following variables entered as co-variates: duration in study (days), baseline dementia severity (CDR SOB), gender of the family carer (male vs female), gender of the person with dementia (male vs female), family carer relationship (spousal vs non-spousal), dementia diagnosis (Alzheimer vs non-Alzheimer dementia), living status (co-resident vs non co-resident), age of the person with dementia and age of the family carer.

Machly's test was used to determine whether the sphericity assumption could be held. Violations in sphericity were checked, and if Prob>ChiSq was greater than or equal to 0.05, then sphericity was assumed. Modifications of degrees of freedom were made upon detecting a

violation in Sphericity. If the epsilon estimate ($\epsilon$) >0.75, we used Hunh-Feldt correction, if $\epsilon$<0.75 then the Greenhouse-Geisser correction was used.

Alongside reporting the main effect of time, statistically significant interactions were also reported alongside their effect size (partial eta, $\eta_p^2$). To assist with the interpretation of these interactions, covariates were entered into appropriate bivariate analysis (t-test, one-way ANOVA), with outcome change scores of the DEMQOL-Proxy and C- DEMOQL (T3-T2 and T2-T1).

Statistical significance was defined as a p value <0.05. All data were analysed in SPSS V.25.

## Results

Two-hundred and forty-eight participants provided data on at least two of the three time-points. On average, 426 days (sd = 203.8 (range 95 to 790) passed between T1 and T2, and 141 days (sd = 22.8 (range 94 to 197) between T2 and T3. The average age of the person with dementia was 77.5 (sd = 8.03) and 69.2 (sd = 10.81) for the family carer. The large majority of people with dementia and family carers were White British/European (99.2% and 99.2% respectively). The family carer was most frequently a spouse/partner (n = 197, 79.4%) of the person with dementia, followed by a son/daughter (n = 49, 19.8%), and the remaining were either a friend or other family member (n = 2, 0.8%). By the end of the study, 77.3% (n = 157) of people with dementia were living with their family carer, 12.3% were living independently, and 10.4% (n = 21) were living in a care home. Of those living in a care home, six had moved into care since March 2020. Full demographic data is presented in Table 1.

### Change in mean scores at each time point

Timeline and change in mean scores for DEMQOL-Proxy and C-DEMQOL are provided in Figs 1 and 2.

### DEMQOL-Proxy

A repeated measures ANOVA with a Huynh-Feldt correction showed that mean DEMQOL-Proxy did not differ significantly between time points [F (2.00, 354.00) = 0.61, p = 0.54, $\eta_p^2$ = 0.003]. Post hoc analysis revealed a small increase in DEMQOL-Proxy scores between T1 and T2 (MD = -0.27, 95% -2.44 to 1.89), and a small decline between T1 and T3 (MD = 0.12, 95% CI = -1.97 to 2.21). Living status (p = 0.02, $\eta_p^2$ = 0.03), carer gender (p = 0.04, $\eta_p^2$ = 0.02) and baseline dementia severity (p = 0.04, $\eta_p^2$ = 0.02) demonstrated a significant effect on DEM-QOL-Proxy scores over time.

Male carers were more likely to report an improvement in DEMQOL-Proxy scores (M = 3.14, SD = 13.99) compared to female carers (M = -1.11, SD = 11.84) between T1 and T2 (MD = 4.26, 95%CI = 0.72 to 7.79, p = 0.02). No other variable was significantly associated with change scores, see Table 2 for further information.

### C-DEMQOL

A repeated measures ANOVA with a Huynh-Feldt correction showed that mean C-DEMQOL did not differ significantly between time points [F (1.99, 359.23) = 0.82, p = 0.44, $\eta_p^2$ = 0.005]. Post hoc tests using Bonferroni correction revealed that the C-DEMQOL reduced by an aver-age of 1.98 points by T2 (95%CI = -0.13 to 4.01) and 3.40 points by T3 (95%CI 1.42 to 5.38). There was a non-statistically significant decline in C-DEMQOL scores between T2 and T3 (MD = 1.42, 95%CI = -0.26 to 3.10). Living status (p<0.001, $\eta_p^2$ = 0.06) and baseline dementia

**Table 1. Baseline participant characteristics (n = 248).**

| | Person with dementia | | Carer | |
|---|---|---|---|---|
| | **M** | **SD** | **M** | **SD** |
| Age | 77.47 | 8.03 | 70.08 | 10.60 |
| | **N** | **%** | **N** | **%** |
| Gender | | | | |
| Male | 145 | 58.5 | 79 | 31.9 |
| Female | 103 | 41.5 | 169 | 68.1 |
| Ethnicity | | | | |
| White British/European | 246 | 99.2 | 245 | 98.8 |
| Mixed/ Multiple Ethnic Groups | 1 | .4 | 0 | 0 |
| Black/African/Caribbean/Black British | 1 | .4 | 2 | 0.8 |
| Missing | 0 | 0 | 1 | 0.4 |
| Current Marital Status | | | | |
| Currently married | 194 | 78.2 | 213 | 85.9 |
| Other marital status (e.g. widow, cohabiting) | 54 | 21.7 | 35 | 14 |
| Highest level of education | | | | |
| Less than primary school | 1 | .4 | 0 | 0 |
| Primary school completed | 14 | 5.6 | 9 | 3.6 |
| Secondary school completed | 101 | 40.7 | 80 | 32.3 |
| Greater than secondary school completed | 129 | 52 | 159 | 64.2 |
| Missing | 3 | 1.2 | 0 | 0 |
| Currently employed | | | | |
| Yes | 7 | 2.8 | 49 | 19.8 |
| No | 240 | 96.8 | 199 | 80.2 |
| Missing | 1 | 0.4 | 0 | 0 |
| Rurality | | | | |
| Urban | - | - | 48 | 19.4 |
| Suburbs | - | - | 109 | 44.0 |
| Rural | - | - | 90 | 36.3 |
| Missing | - | - | 1 | 0.4 |
| Living status | | | | |
| Non co-resident | | | 91 | 36.7 |
| Co-resident | | | 157 | 63.3 |
| Relationship to the person with dementia | | | | |
| Spouse/partner | - | - | 197 | 79.4 |
| Son/Daughter | - | - | 49 | 19.8 |
| Other | - | - | 2 | 0.8 |
| Diagnosis | | | | |
| Alzheimer's Disease | 93 | 37.5 | - | - |
| Mixed (for all mixed diagnosis) | 66 | 26.6 | - | - |
| Vascular | 37 | 14.9 | - | - |
| Other | 49 | 19.6 | - | - |
| Missing | 3 | 1.2 | - | - |
| CDR | | | | |
| 0 | 0 | 0 | - | - |
| 0.5 (Questionable) | 50 | 20.2 | - | - |
| 1.0 (Mild) | 106 | 42.7 | - | - |
| 2.0 (Moderate) | 73 | 29.4 | - | - |

(*Continued*)

**Table 1.** (Continued)

| | Person with dementia | | Carer | |
|---|---|---|---|---|
| | **M** | **SD** | **M** | **SD** |
| 3.0 (Severe) | 11 | 4.4 | - | - |
| Missing | 8 | 3.2 | - | - |

severity (p = 0.002, $\eta_p^2$ = 0.03) demonstrated a significant effect on C-DEMQOL scores over time.

For baseline dementia severity, having milder severity was a predictor of greater C-DEMQOL decline between T1 and T2 compared to more severe participants (F = 6.45, p<0.001). C-DEMQOL scores were more likely to decline between T1 and T2 in co-resident carers than those that did not (MD = 7.11, 95%CI = 3.53 to 10.69, p <0.001). Dementia severity and living

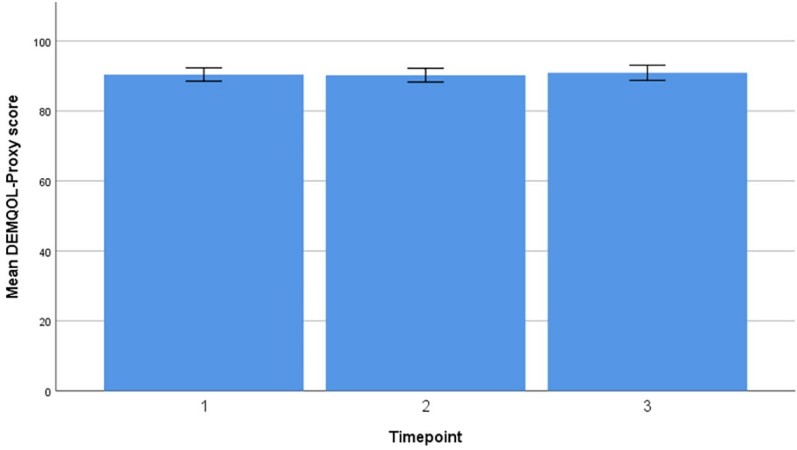

**Fig 1. Timeline and change in mean scores (DEMQOL-Proxy).**

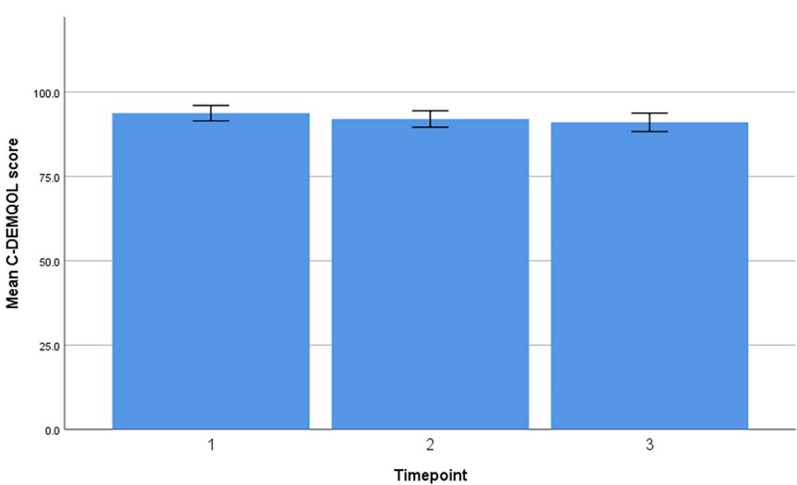

**Fig 2. Timeline and change in mean scores (C-DEMQOL).**

**Table 2. Bivariate comparisons of DEMQOL-Proxy (change between time-points).**

| | T2-T1 | | | T3-T2 | | |
|---|---|---|---|---|---|---|
| | **Mean** | **SD** | **Co-eff (p)** | **Mean** | **SD** | **Co-eff (p)** |
| Living status | | | T = 1.54 (p = 0.13) | | | T = -0.32 (p = 0.75) |
| Non co-resident | 1.93 | 13.80 | | -0.83 | 9.22 | |
| Co-resident | -0.76 | 11.94 | | -0.28 | 10.01 | |
| Carer gender | | | T = 2.37 (p = 0.02) | | | T = -1.11 (p = 0.27) |
| Male | 3.14 | 13.99 | | -1.63 | 10.86 | |
| Female | -1.11 | 11.84 | | 0.10 | 9.36 | |
| Severity Baseline (CDR) | | | F = 0.38 (p = 0.77) | | | F = 0.86 (p = 0.47) |
| Questionable | -1.08 | 12.39 | | -2.45 | 10.15 | |
| Mild | .81 | 12.52 | | -.15 | 10.10 | |
| Moderate | .45 | 13.12 | | .80 | 9.61 | |
| Severe | -2.53 | 13.48 | | -.56 | 6.43 | |

status were not significantly associated with C-DEMQOL change scores between T2 and T3. See Table 3.

**C-DEMQOL subscales.** The 'meeting person needs' subscale did not significantly differ between time points F (2.00, 366.00) = 1.04, p = 0.36, $\eta_p^2$ = 0.006]. Post hoc analysis revealed a 0.86-point decline by T2 (95%CI 0.17 to 1.54), but no significant difference between T2 and T3 (MD = 0.34, 95%CI -0.27 to 0.95). Significant interaction effects were detected for relationship status (p = 0.04, $\eta_p^2$ = 0.02), baseline CDR (p<0.001, $\eta_p^2$ = 0.06) and living status (p<0.001, $\eta_p^2$ = 0.10).

A repeated measures ANOVA with a Huynh-Feldt correction showed that the 'carer wellbeing' subscale did not significantly differ between time points F (2.00, 369.07) = 0.57, p = 0.57, $\eta_p^2$ = 0.003]. Post hoc analysis revealed a 0.85-point decline on the subdomain between T1 and T3 (95%CI 0.28 to 1.42), though there were no significant differences between any other timepoints. Significant interaction effects were detected for baseline CDR only (p = 0.009, $\eta_p^2$ = 0.03).

The 'patient relationship' subscale did not significantly differ between time points F (2, 350) = 1.44, p = 0.24, $\eta_p^2$ = 0.01. Post hoc analysis revealed that there was a statistically significant decline at T2 (MD = 0.67, 95%CI 0.15 to 1.18) but no significant difference between T2 and T3 (MD = 0.13, 95%CI—0.35 to 0.62). There were no interaction effects with any covariates.

A repeated measures ANOVA with a Huynh-Feldt correction highlighted that the 'confidence in future' subscale did not significantly differ between time points F (2.00, 364.00) =

**Table 3. Bivariate comparisons of C-DEMQOL (change between time-points).**

| | T2-T1 | | | T3-T2 | | |
|---|---|---|---|---|---|---|
| | **M** | **SD** | **Co-eff (p)** | **M** | **SD** | **Co-eff (p)** |
| Living status | | | T = 4.17 (p<0.001) | | | T = 0.36 (p = 0.72) |
| Non co-resident | 2.98 | 14.29 | | -0.02 | 0.62 | |
| Co-resident | -4.13 | 11.48 | | -0.06 | 0.57 | |
| Severity Baseline (CDR) | | | F = 6.45 (p<0.001) | | | F = 0.50 (p = 0.69) |
| Questionable | -4.14 | 13.20 | | -2.59 | 8.72 | |
| Mild | -4.09 | 12.40 | | -1.54 | 9.14 | |
| Moderate | 2.41 | 11.75 | | -.91 | 12.45 | |
| Severe | 7.65 | 16.14 | | 2.22 | 6.24 | |

0.72, p = 0.49, $\eta_p^2$ = 0.004. Between T1 and T2 there was a decline of 0.16 points (95%CI -0.45 to 0.76) and between T2 and T3 there was decline of 0.13 points (95%CI -0.46 to 0.72), neither were statistically significant Only living status (p = 0.02, $\eta_p^2$ = 0.02) had a significant effect on the confidence in future subscale over time.

The 'feeling supported' subscale did not significantly differ between time points, F(2.00, 306.00) = 1.24, p = 0.29, $\eta_p^2$ = 0.008. Scores did not significantly change between T1 and T2 (MD = -0.41, 95% CI -1.20 to 0.38). However, there was a statistically significant 0.86 decline between T2 and T3 (95%CI 0.14 to 1.58). Again, living status had a significant effect on this subscale over time (p = 0.006, $\eta_p^2$ = 0.03).

## Discussion

This study set out to explore the effect of the COVID-19 pandemic and subsequent restrictions on the QoL of a cohort of people with dementia and their family carers. Overall, it is striking that people with dementia and their carers did not report a decline in QoL during the pandemic or in the months following restrictions. However, there was variation in subgroups and our data suggest that family carers co-resident with the person with dementia they care for were particularly strongly affected in terms of having the greatest decrease in QoL.

We observed that the QoL of people with dementia was generally stable over the three time points, but there was an association with carer gender. Bivariate analysis indicated that QoL was more likely to improve for those with male carers between the first and second timepoint. It is unclear why the gender of the carer should affect change of QoL of the person with dementia. Theoretically, it could be attributed to gendered differences between family carers in how they have handled care and support due to the pandemic, with taking charge being seen as a key coping and adjustment process [18] perhaps to the benefit of the person they care for. Alternatively, it could be that female carers, due to their gender would have been more likely [19, 20] to have been delivering more hands-on care before the pandemic, and as such, might have been more physically and emotionally burdened pre-pandemic, compared to their male counterparts, and therefore less likely to experience any improvements in QoL during the pandemic. However, it may also reflect gender differences in reporting, with males adopting a 'stiff upper lip' mentality associated with masculinity [21] or over-estimating QoL of the care recipient as a means to boost masculine status [22] and not to admit failure. Further research is needed to understand the influence of masculinity and gender-related coping strategies on the QoL of people with dementia.

For carer QoL, there was not a decline over the study period. Living status and dementia severity were the only factors to have an effect on carer QoL scores within the multivariate model. Bivariate analysis highlighted that co-resident carers and carers who support the least impaired, experienced the greatest decline in QoL. This association was most notable between the first two timepoints.

Cognitive status and co-resident status remained associated with change scores. Initially, it may seem counter-intuitive that greater cognitive impairment may be associated with improving carer QoL from before the pandemic to the first lockdown. However, it is likely that those with higher cognitive function at baseline had the propensity to experience more cognitive decline than those who were already relatively impaired. As such, carer QoL may represent the impact and adjustment of caring for someone with increasingly impaired cognition. The reduction in services and social restrictions during the pandemic will also have meant that for co-resident carers, there would have been increased time spent together, therefore possibly decreasing carers' time for respite. Additionally, it is likely the lockdown restrictions would have diminished the ability of carers to be able to undertake activities which offer personal

meaning, a factor known to influence carer QoL [23, 24] In the C-DEMQOL development, personal freedom and carer independence were seen as an important component of carer QoL [15, 25, 26], which may have been lost as a result of the pandemic restrictions. Even when exploring these associations on the C-DEMQOL subscales, either dementia severity and/or living status consistently demonstrated interaction effects, with the exception of the patient relationship subscale. The persistent negative effect of co-residence over the two phases of the study underscores its importance in carer QoL terms.

Whilst there can be no comparison group with a pandemic, historical data suggest that DEMQOL-Proxy scores are relatively stable over longer durations before the pandemic (18 months; MD = 1.56) [27]. Our data suggest that overall, the pandemic and associated restrictions may have had less of an impact on people's QoL than perhaps we expected, or possibly that an adaptation process has occurred to a chronic stress, namely living with dementia for the person with the diagnosis and for their carer.

It might also be that the social restrictions associated with the pandemic have been less impactful due to pre-pandemic reduction in social networks and associated impact on QoL [28].

However, at a more granular level there were subscales of the C-DEMQOL that did change between specific timepoints. From the pre-pandemic period to the first lockdown, our data suggest that meeting personal needs and the carer-patient relationship declined. This perhaps highlights the impact of adjusting to relatively recent restrictions in terms of the halting of support services and shielding. During the pandemic, from the first to the second lockdown there was no recovery in these subscales but they did not continue to decline. But in this second phase there were new declines in the C-DEMQOL feeling supported subscales. These may represent longer term effects of the pandemic on carer QoL.

Our findings suggest that services need to proactively identify people with dementia and their carers who are more likely to be struggling due to the pandemic. The need for services to adapt and provide targeted support to more vulnerable groups of carers is of upmost importance.

There are important limitations to this study. First, as an observational study, we are only able to report associations and cannot make causal inferences. Whilst this study took place in the context of COVID-19 there are other possible reasons for changes in the QoL of people with dementia and their carers which were not measured in this study. Second, the sample was composed of predominantly White British and spousal carers, which may limit consideration of changes by ethnic group and in non-spousal caregivers. The sample was also recruited from England and caution is needed generalising findings to other countries where dementia services and support may have been differentially affected by government restrictions. Third, this is an opportunistic study and baseline data were collected at the time of recruitment into TFD, so the time that passed between baseline measurement and the start of the pandemic was variable and substantial. Fourth, QoL of the person with dementia was assessed using an informant report, and not self-report, whilst the two perspectives have high agreement, it is possible the informant report introduces different biases. The DEMQOL-Proxy does however allow us to estimate the QoL of people with dementia across all severity groups, rather than those who just have mild to moderate dementia. Alongside these limitations there are strengths in the size of the group studied, the two assessment time points during the pandemic and the use of well validated quantitative measures of QoL for the person with dementia and the family carer.

This study provides useful insights into the potential impact of the COVID-19 pandemic and associated public health restrictions on the QoL of people with dementia and their family carers. Whilst the nature of the study limits inference, the study provides novel evidence that, at a group level, the QoL of people with dementia remained stable over the pandemic. In

contrast, for co-resident family carers there was a consistent decline in QoL through the pandemic. Given their central importance in supporting people with dementia and the negative clinical and economic consequences of carer burnout, our data highlights the importance of effective support services for carers of people with dementia.

## Acknowledgments

We would like to thank the following staff for their contribution to the study; Dr Sam Robertson, Rachel Russell, Natalie Portwine, Tamsin Eperson, Alice Russell, Jacob Reichental, Anomita Karim, Marcela Carvajal, Bethany Fine.

We would also to like to thank the contribution of the members of the lived experience advisory group; Julia Fountain, Ellen Jones, Claire Parker, Fiona McGhee, Tina and Ian Wakeford.

## Author Contributions

**Conceptualization:** Stephanie Daley, Sube Banerjee.

**Data curation:** Stephanie Daley, Elise Armsby, Nazire Akarsu, Joanna Pooley, Georgia Towson, Yvonne Feeney, Bethany Fine.

**Formal analysis:** Nicolas Farina, Laura Hughes.

**Funding acquisition:** Stephanie Daley, Naji Tabet, Sube Banerjee.

**Investigation:** Sube Banerjee.

**Methodology:** Stephanie Daley, Nicolas Farina, Laura Hughes, Sube Banerjee.

**Project administration:** Stephanie Daley, Joanna Pooley, Yvonne Feeney.

**Resources:** Naji Tabet.

**Supervision:** Stephanie Daley, Joanna Pooley, Yvonne Feeney, Sube Banerjee.

**Writing – original draft:** Stephanie Daley, Nicolas Farina, Joanna Pooley, Bethany Fine, Sube Banerjee.

**Writing – review & editing:** Stephanie Daley, Nicolas Farina, Laura Hughes, Elise Armsby, Nazire Akarsu, Georgia Towson, Yvonne Feeney, Naji Tabet, Sube Banerjee.

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
