## [Decision Letter · Decision Letter 0]

4 Aug 2021

PONE-D-21-19534

Covid-19 and the quality of life of people with dementia and their carers – the TFD-C19 Study

PLOS ONE

Dear Dr. Daley,

Thank you for submitting your manuscript to PLOS ONE. After careful consideration by 2 Reviewers and an Academic Editor, all of the critiques of Reviewer #1 must be addressed in detail in a revision to determine publication status, especially the issues surrounding statistical analyses. If you are prepared to undertake the work required, I would be pleased to reconsider my decision, but revision of the original submission without directly addressing the critiques of Reviewer #1 does not guarantee acceptance for publication in PLOS ONE. If the authors do not feel that the queries can be addressed, please consider submitting to another publication medium. A revised submission will be sent out for re-review. The authors are urged to have the manuscript given a hard copyedit for syntax and grammar.

“This study was funded by the National Institute for Health Research (NIHR) Applied Research Collaboration Kent, Surrey, Sussex (grant number N/A), as well as Sussex Partnership NHS Foundation Trust and Brighton & Sussex Medical School”.

 However, funding information should not appear in the Acknowledgments section or other areas of your manuscript. We will only publish funding information present in the Funding Statement section of the online submission form.

 “This study was funded by the National Institute for Health Research (NIHR) Applied Research Collaboration Kent, Surrey, Sussex (grant number N/A), as well as Sussex Partnership NHS Foundation Trust and Brighton & Sussex Medical School The funders had no role in study design, data collection and analysis, decision to publish, or preparation of the manuscript”.

**Comments to the Author**

1. Is the manuscript technically sound, and do the data support the conclusions?

Reviewer #1: Partly

Reviewer #2: Yes

2. Has the statistical analysis been performed appropriately and rigorously? 

Reviewer #1: No

Reviewer #2: Yes

3. Have the authors made all data underlying the findings in their manuscript fully available?

Reviewer #1: No

Reviewer #2: No

4. Is the manuscript presented in an intelligible fashion and written in standard English?

Reviewer #1: Yes

Reviewer #2: Yes

5. Review Comments to the Author

Reviewer #1: In the manuscript PONE-D-21-19534, the Authors investigated possible changes in quality of life of individuals with dementia and their careers during COVID-19 pandemic. Interestingly, the Authors had the opportunity to assess this psychological component twice, and to compare it with data collected before the pandemic (i.e., the baseline). Results suggested no substantial changes in quality of life in the assessed population. This results are unexpected in some way, but also they might be very interesting: why did not these groups of people report any changes in their quality of life, in a period in which health services were not approachable as well as the rest of populations suffered from psychological distress? This intriguing results might offer some consideration about what the level (maybe very low) of quality of life of these people before the pandemic. Even though this manuscript sounds very interesting, some methodological criticisms (and specifically in terms of statistical analyses) did not allow me to express a positive feedback about it.

Specifically, the Authors performed two independent analyses in which they compared T0 vs T1, and T1 vs T2, without a clear explanation relative to this choice (the Authors reported “due to the temporal differences” as reason to perform two t-tests, but again this is not a clear explanation). I would suggest to use a repeated measure ANOVA in which the comparisons can be performed between the three time points (to, t1 and t2) in the same analysis. If the Authors performed a repeated measure ANOVA, they can introduce demographical information as well as any other factors as covariates. This analysis might sound more in line with the aim of the manuscript (changes in QoL during the pandemic).

My second concern regarded the use of the regression analyses to verify the role of the demographical factors on the main differences. The use of the regression analyses should be carefully considered, because the Authors used demographical information (in other words, no controlled factor) as predictors. Since the nature of their data, a multivariate analyses might be more adapted (as I have previously suggested). Moreover, the use of this analyses was in disagreement with what reported in the limitation of study (linea 299-300): indeed, regression analyses are meant to described inferential relationship. Because of this criticism, the results relative to the role of the demographical components on quality of life cannot be correctly understood.

In revised their paper, I strongly recommend the Authors to:

- clarify the concept of quality of life (lines 34-35), since it represents a crucial point in their work ; moreover in lines 35-37, the Authors reported that the previous studies did not adopt well-validated measurements? And if it is the case, why it should be matter?

- clarify the aim of the study and predictions, at the end of the Introduction;

- improve tables in terms of clarity, avoiding acronyms and underlining significant differences. Also, the order was altered between table 5 and 6;

- use graphs to show the differences between the scores reported across the three time points; also, a graphical representation of the time points in relation to the contextual information might help Readers in understanding the timeline;

- devote more effort in describing the questionnaires used for their research. For example, what dimensions are measured by DEMQOL_Proxy? Is there any information about statistical validity of this questionnaire? The same information should be reported for all the other measurements. Moreover, what is the difference between the DEMQOL_Proxy and the C_DEMQOL? Moreover, when the CDS was administered (t0 or t1)?

- in describing the results, measures relative to the effect size should be reported. Moreover, I encourage the authors to report 95 % CI in their analysis, since no a priori sample size was performed (and this is completely understandable).

- devote more effort in describing the clinical and social implications from their study. Perhaps, people with dementia and their families live a kind of “self-isolation”, which is not so amplified by the pandemic social restriction.

Reviewer #2: This was a nested study within a larger ongoing study to assess changes (initial change due to pandemic and ongoing change during the pandemic) in mainly the quality of life (QoL) of people with dementia and their carers during the pandemic lockdown in England. The study found that there was no overall change in QoL but certain subgroups were associated with changes in QoL subscale items and in overall QoL scores.

The study used validated instruments and and made comparisons with pre-pandemic scores, as well as between two time points during the pandemic, which were strengths of the study, as many COVID-19 studies on dementia wellbeing are cross sectional in design.

The article is clearly written and presented. Limitations were adequately described.

6. PLOS authors have the option to publish the peer review history of their article (what does this mean?). If published, this will include your full peer review and any attached files.

**Do you want your identity to be public for this peer review?** For information about this choice, including consent withdrawal, please see our Privacy Policy.

Reviewer #1: No

Reviewer #2: No

We look forward to receiving your revised manuscript.

Kind regards,

Stephen D. Ginsberg, Ph.D.

Section Editor

PLOS ONE

---

## [Author Response · Author response to Decision Letter 0]

8 Dec 2021

Amendment requested Changed made 

Thank you for this, we have amended the format to follow PLOS ONE style requirements

Thank you for stating the following in the Acknowledgments Section of your manuscript:

“This study was funded by the National Institute for Health Research (NIHR) Applied Research Collaboration Kent, Surrey, Sussex (grant number N/A), as well as Sussex Partnership NHS Foundation Trust and Brighton & Sussex Medical School”.

 However, funding information should not appear in the Acknowledgments section or other areas of your manuscript. We will only publish funding information present in the Funding Statement section of the online submission form.

 “This study was funded by the National Institute for Health Research (NIHR) Applied Research Collaboration Kent, Surrey, Sussex (grant number N/A), as well as Sussex Partnership NHS Foundation Trust and Brighton & Sussex Medical School The funders had no role in study design, data collection and analysis, decision to publish, or preparation of the manuscript”. Please include your amended statements within your cover letter; we will change the online submission form on your behalf. We have remoted the funding text from the main document

There is no change to our funding statement and thank you for making the necessary changes.

Please include captions for your Supporting Information files at the end of your manuscript, and update any in-text citations to match accordingly. Please see our Supporting Information guidelines for more information: http://journals.plos.org/plosone/s/supporting-information.

Captions for supporting information files have been added 

The PLOS Data policy requires authors to make all data underlying the findings described in their manuscript fully available without restriction, with rare exception (please refer to the Data Availability Statement in the manuscript PDF file). The data should be provided as part of the manuscript or its supporting information, or deposited to a public repository. For example, in addition to summary statistics, the data points behind means, medians and variance measures should be available. If there are restrictions on publicly sharing data—e.g. participant privacy or use of data from a third party—those must be specified. Thank you in terms of data sharing, Consent was not obtained from participants to share data online. Data will be made available upon reasonable request to the corresponding author

Specifically, the Authors performed two independent analyses in which they compared T0 vs T1, and T1 vs T2, without a clear explanation relative to this choice (the Authors reported “due to the temporal differences” as reason to perform two t-tests, but again this is not a clear explanation). I would suggest to use a repeated measure ANOVA in which the comparisons can be performed between the three time points (to, t1 and t2) in the same analysis. If the Authors performed a repeated measure ANOVA, they can introduce demographical information as well as any other factors as covariates. This analysis might sound more in line with the aim of the manuscript (changes in QoL during the pandemic) Thank you for this useful suggestion. In a major revision we have amended the analyses following the reviewers recommendations. This is detailed in methods (ln 140-167) and findings. 

Summary statistics were generated on demographic information for the person with dementia and family carer. Means and standard deviations were reported for continuous data, frequencies and percentages were reported for categorical data. Categorical data were dichotomised. Total scores for the C-DEMQOL (and its subscales) and DEMQOL-Proxy were calculated according to their original development guidelines. This included the rules for handling missing data.

To understand how QoL changed over time, a repeated measures ANOVA was run for each dependent variable with the following variables entered as co-variates: duration in study (days), baseline dementia severity (CDR SOB), sex of the family carer (male vs female), sex of the person with dementia (male vs female), family carer relationship (spousal vs non-spousal), dementia diagnosis (Alzheimer vs non-Alzheimer dementia), living status (co-resident vs non co-resident), age of the person with dementia and age of the family carer.

Machly’s test was used to determine whether the sphericity assumption could be held. Violations in sphericity were checked, and if Prob>ChiSq was greater than or equal to 0.05, then sphericity was assumed. Modifications of degrees of freedom were made upon detecting a violation in Sphericity. If the epsilon estimate (ε) >0.75, we used Hunh-Feldt correction, if ε<0.75 then the Greenhouse-Geisser correction was used. 

Alongside reporting the main effect of time, statistically significant interactions were also reported alongside their effect size (partial eta, ηp2). To assist with the interpretation of these interactions, covariates were entered into appropriate bivariate analysis (t-test, one-way ANOVA), with outcome change scores of the DEMQOL-Proxy and C- DEMOQL (T3-T2 and T2-T1). 

My second concern regarded the use of the regression analyses to verify the role of the demographical factors on the main differences. The use of the regression analyses should be carefully considered, because the Authors used demographical information (in other words, no controlled factor) as predictors. Since the nature of their data, a multivariate analyses might be more adapted (as I have previously suggested). Moreover, the use of this analyses was in disagreement with what reported in the limitation of study (linea 299-300): indeed, regression analyses are meant to described inferential relationship. Because of this criticism, the results relative to the role of the demographical components on quality of life cannot be correctly understood. Thank you again. We have amended the analyses (ln 140-167) to follow the reviewers recommendations. This is detailed in methods and findings. 

Clarify the concept of quality of life (lines 34-35), since it represents a crucial point in their work ; moreover in lines 35-37, the Authors reported that the previous studies did not adopt well-validated measurements? And if it is the case, why it should be matter? Thank you for this point, we have added in a definition of quality of life. 

QoL can be defined as the subjective assessment of the position of the individual in their life, in the context to their culture and value system, relating to their goals, expectations and concerns, impacted by their physical and psychological health, level of independence, and social relationships (10). (Ln 37-40)

We have also amended the text to make clearer that there is a need for condition-specific measures of quality in life in dementia, rather than there being a problem with validated measures per se

Living with, or caring for somebody with dementia impacts upon all of the factors known to affect QoL, it is likely that Covid-19 and associated restrictions may have been more impactful for this group of people. There is therefore a need to measure the impact of Covid-19, using condition-specific measures of QoL, which are tailored to the challenges and realities of daily life with dementia. (Ln 40-44)

Clarify the aim of the study and predictions, at the end of the Introduction; Thank you for this point, the study aim has been added 

Our study aim was to understand how COVID-19 had affected the QoL, wellbeing, and care of people with dementia and their family carers. (Ln 56-57)

Improve tables in terms of clarity, avoiding acronyms and underlining significant differences. Also, the order was altered between table 5 and 6; All tables have been improved in terms of clarity and the order is correct 

Use graphs to show the differences between the scores reported across the three time points; also, a graphical representation of the time points in relation to the contextual information might help Readers in understanding the timeline; This information has been added into the manuscript and is shown in Fig 1 and Fig 2.

Devote more effort in describing the questionnaires used for their research. For example, what dimensions are measured by DEMQOL_Proxy? Is there any information about statistical validity of this questionnaire? The same information should be reported for all the other measurements. Moreover, what is the difference between the DEMQOL_Proxy and the C_DEMQOL? 

Moreover, when the CDR was administered (t0 or t1)? 

 Thank you for this point, the CDR was administered at all time points. The text has been amended to make clear that all measures were used at each time point, and also the provide more information on each of the measures – eg domains and psychometric properties.(Ln 104-132)

The difference between DEMQOL-Proxy and C-DEMQOL has been made clear in the next; namely that DEMQOL-Proxy is a proxy measure of QoL in the person with dementia (as reported by carer) and C-DEMQOL is a self-report of QoL in family carers of people with dementia (Ln 128)

In describing the results, measures relative to the effect size should be reported. Moreover, I encourage the authors to report 95 % CI in their analysis, since no a priori sample size was performed (and this is completely understandable). 

 Thank you for this point, this has been amended

Devote more effort in describing the clinical and social implications from their study. Perhaps, people with dementia and their families live a kind of “self-isolation”, which is not so amplified by the pandemic social restriction Thank you for this suggestions, we have expanded on this point within the manuscript (Ln 319-435)

It might also be that the social restrictions associated with the pandemic have been less impactful due to pre-pandemic reduction in social networks and associated impact on QoL (29). 

However, at a more granular level there were subscales of the C-DEMQOL that did change between specific timepoints. From the pre-pandemic period to the first lockdown, our data suggest that meeting personal needs and the carer-patient relationship significantly declined. This perhaps highlights the impact of adjusting to relatively recent restrictions in terms of the halting of support services and shielding. During the pandemic, from the first to the second lockdown there was no recovery in these subscales but they did not continue to decline. But in this second phase there were new declines in the C-DEMQOL feeling supported subscales. These may represent longer term effects of the pandemic on carer QoL. 

Our findings suggest that services need to proactively identify people with dementia and their carers who are more likely to be struggling due to the pandemic. The need for services to adapt and provide targeted support to more vulnerable groups of carers is of upmost importance. (Ln 419-435)

---

## [Decision Letter · Decision Letter 1]

14 Dec 2021

PONE-D-21-19534R1Covid-19 and the quality of life of people with dementia and their carers – the TFD-C19 StudyPLOS ONE

Dear Dr. Daley,

Thank you for resubmitting your work to PLOS ONE. Please make the corrections posed by Reviewer #1 so I can render a decision on this manuscript.

**Comments to the Author**

1. If the authors have adequately addressed your comments raised in a previous round of review and you feel that this manuscript is now acceptable for publication, you may indicate that here to bypass the “Comments to the Author” section, enter your conflict of interest statement in the “Confidential to Editor” section, and submit your "Accept" recommendation.

Reviewer #1: All comments have been addressed

2. Is the manuscript technically sound, and do the data support the conclusions?

Reviewer #1: Yes

3. Has the statistical analysis been performed appropriately and rigorously? 

Reviewer #1: Yes

4. Have the authors made all data underlying the findings in their manuscript fully available?

Reviewer #1: Yes

5. Is the manuscript presented in an intelligible fashion and written in standard English?

Reviewer #1: Yes

6. Review Comments to the Author

Reviewer #1: Here, I reported my review about the revised version of the article PONE-D-21-19534R1, entitled "Covid-19 and the quality of life of people with dementia and their carers – the TFD-C19 Study". The Authors made a remarkable work in revising the manuscript, especially in the statistical and result sections.

I have only one main concern, that regards the CDR and SF-DEM questionnaires. Indeed, in the revised version of the manuscript, I was not able to find the data about CDR and SF-DEM. Indeed, only the results relative to the QoL – related questionnaires were clearly reported in the Results; so that, the Authors should clarify this point. Also, figures for these two questionnaires may be used to show the scores in the different time points.

Minor points:

• Abstract seemed to be too long, and not in line with the Journal’s style.

• CDR and SF-DEM should not be described along the two QoL – related questionnaires (i.e., the main outcomes of the study) in the Methods, because they assessed two different components.

• Data about the two QoL-related questionnaire would be shown in different figures, since they were related to different scores, with different minimum/maximum, as well as to two different components. Also, the Authors should devote more effort in describing figures in the captions. Such as: “In Figure X, mean (bar) and standard deviation (vertical line) about the mean score (y-axis) registered in thee time point (T0 = before the pandemic; T1 = after XX months after the pandemic; and T2 = XX) at the XXX questionnaire is shown”.

• Pag. 14, lines 270-272: this sentence seemed to be not entirely clear; please, rephrase it.

• In the Discussion, terms like “significantly different” should be avoided.

• I am not a native speaker, but I am wondering if gender rather than sex may be more appropriate as label in the manuscript, such as "carer sex" may be "carer gender".

7. PLOS authors have the option to publish the peer review history of their article (what does this mean?). If published, this will include your full peer review and any attached files.

**Do you want your identity to be public for this peer review?** For information about this choice, including consent withdrawal, please see our Privacy Policy.

Reviewer #1: No

We look forward to receiving your revised manuscript.

Kind regards,

Stephen D. Ginsberg, Ph.D.

Section Editor

PLOS ONE

---

## [Author Response · Author response to Decision Letter 1]

22 Dec 2021

Amendment requested Changed made 

I have only one main concern, that regards the CDR and SF-DEM questionnaires. Indeed, in the revised version of the manuscript, I was not able to find the data about CDR and SF-DEM. Indeed, only the results relative to the QoL – related questionnaires were clearly reported in the Results; so that, the Authors should clarify this point. Thank you for this very helpful suggestion. 

We have removed all of the text related to the SF-DEM (ln 119-124) from the manuscript as you rightly point out it, it is not reflected in the aims or results of the study

We have however kept in the text related to the CDR, as the baseline CDR measurement was used to assess whether dementia severity was associated with change in QoL scores over time. This is also reported in the results section in Table 1: Baseline participant characteristics (n=248)

However, we have moved the CDR related text to the end of the measure section. We have also made the purpose of the using the CDR clearer (as below): 

This measure was used to assess whether dementia severity was associated with change in QoL scores over time (ln 140-141) 

Also, figures for these two questionnaires may be used to show the scores in the different time points. We have deleted the original Figure 1 (timescales) and replaced two condensed figures. The first (F1) for C-DEMQOL-Proxy with dates and change in mean scores, and the second (F2) for C-DEMQOL with dates and change in mean scores

Abstract seemed to be too long, and not in line with the Journal’s style. We have reviewed the journal’s guidance on abstracts (namely 300 words, explain how study was done, summarise most important results and their significance)

The abstract was 306 words, and we have now reduced this to 300 words. We believe that the abstract does follow the journal’s guidance.

CDR and SF-DEM should not be described along the two QoL – related questionnaires (i.e., the main outcomes of the study) in the Methods, because they assessed two different components. As discussed above, we have removed the SF-DEM and have altered the text on the CDR

Data about the two QoL-related questionnaire would be shown in different figures, since they were related to different scores, with different minimum/maximum, as well as to two different components. Also, the Authors should devote more effort in describing figures in the captions. Such as: “In Figure X, mean (bar) and standard deviation (vertical line) about the mean score (y-axis) registered in thee time point (T0 = before the pandemic; T1 = after XX months after the pandemic; and T2 = XX) at the XXX questionnaire is shown We have amended these two figures (F1, F2) as requested. 

Pag. 14, lines 270-272: this sentence seemed to be not entirely clear; please, rephrase it. Our apologies for this. We have re-worded as follows:

Between T1 and T2 there was a decline of 0.16 points (95%CI -0.45 to 0.76) and between T2 and T3 there was decline of 0.13 points (95%CI -0.46 to 0.72), neither were statistically significant (ln 260-262)

 In the Discussion, terms like “significantly different” should be avoided.

 We have removed this term throughout the discussion. 

I am not a native speaker, but I am wondering if gender rather than sex may be more appropriate as label in the manuscript, such as "carer sex" may be "carer gender".

 Thank you for highlighting this point, we have changed this term throughout the manuscript

---

## [Editor Report · Decision Letter 2]

27 Dec 2021

Covid-19 and the quality of life of people with dementia and their carers – the TFD-C19 Study

PONE-D-21-19534R2

Dear Dr. Daley,

We’re pleased to inform you that your manuscript has been judged scientifically suitable for publication and will be formally accepted for publication once it meets all outstanding technical requirements.

Kind regards,

Stephen D. Ginsberg, Ph.D.

Section Editor

PLOS ONE

---

## [Editor Report · Acceptance letter]

6 Jan 2022

PONE-D-21-19534R2 

Covid-19 and the quality of life of people with dementia and their carers – the TFD-C19 Study 

Dear Dr. Daley:

I'm pleased to inform you that your manuscript has been deemed suitable for publication in PLOS ONE. Congratulations! Your manuscript is now with our production department. 

Kind regards, 

on behalf of

Dr. Stephen D. Ginsberg 

Section Editor

PLOS ONE